# Variant-specific priors clarify colocalisation analysis

**Jeffrey M. Pullin**[1]*, **Chris Wallace**[1,2]*

**1** MRC Biostatistics Unit, University of Cambridge, Cambridge, United Kingdom, **2** Cambridge Institute of Therapeutic Immunology and Infectious Disease, University of Cambridge, Cambridge, United Kingdom

\* jp2045@cam.ac.uk (JMP), cew54@cam.ac.uk (CW)

## Abstract

Linking GWAS variants to their causal gene and context remains an ongoing challenge. A widely used method for performing this analysis is the coloc package for statistical colocalisation analysis, which can be used to link GWAS and eQTL associations. Currently, coloc assumes that all variants in a region are equally likely to be causal, despite the success of fine-mapping methods that use additional information to adjust their prior probabilities. In this paper we propose and implement an approach for specifying variant-specific prior probabilities in the coloc method. We describe and compare six source of information for specifying prior probabilities: non-coding constraint, enhancer-gene link scores, the output of the PolyFun method and three estimates of eQTL–TSS distance densities. Using simulations and analysis of ground-truth pQTL–eQTL colocalisations we show that variant-specific priors, particularly the eQTL–TSS distance density priors, can improve colocalisation performance. Furthermore, across GWAS–eQTL colocalisations variant-specific priors changed colocalisation significance in up to 14.1% of colocalisations, at some loci revealing the likely causal gene.

**Data availability statement:** The eQTLGen data was downloaded from https://molgenis26.gcc.rug.nl/downloads/eqtlgen/cis-eqtl/2019-12-11-cis-eQTLsFDR0.05-ProbeLevel-CohortInfoRemoved-BonferroniAdded.txt.gz.
The OneK1K data was downloaded from

## Author summary

Evaluating whether two traits, such as disease risk and gene expression, are affected by the same genetic variants is crucial for understanding the molecular mechanisms through which genetic variants act. A widely used method for determining whether there is a shared cause—termed 'colocalisation'—is 'coloc', which updates prior knowledge about the chance of two traits sharing a causal variant with observed genetic association data in a Bayesian statistical framework. However, currently coloc assumes that all variants are equally likely *a priori*—before looking at the data—to affect each trait. This assumption is made despite the existence of considerable information about the different types of variants and large amounts of data linking variants and traits. Here, we describe an approach for incorporating this variant-specific prior information into coloc. We describe several sources of prior information about variants and assess what impact they have on the performance of coloc. We find that some sources of information

https://onek1k.s3.ap-southeast-2.amazonaws.com/esnp/esnp_table.tsv.gz. The eQTL catalogue v6 data (including GTEx v8) was downloaded from the eQTL Catalogue FTP site using the links specified in https://raw.githubusercontent.com/eQTL-Catalogue/eQTL-Catalogue-resources/master/tabix/tabix_ftp_paths.tsv. The FinnGen GWAS summary statistics were downloaded following the FinnGen consortium's instructions https://www.finngen.fi/en/access_results. The UK Biobank summary statistics were downloaded from https://pheweb.org/UKB-SAIGE/: https://pheweb.org/UKB-SAIGE/download/250.2, https://pheweb.org/UKB-SAIGE/download/401, https://pheweb.org/UKB-SAIGE/download/244. Baseline annotations used in the calculation of trait-specific PolyFun prior weights were downloaded from https://broad-alkesgroup-ukbb-ld.s3.amazonaws.com/UKBB_LD/baselineLF_v2.2.UKB.polyfun.tar.gz. The Open Targets Genetics data was downloaded from the Open Targets Genetics FTP site https://ftp.ebi.ac.uk/pub/databases/opentargets/genetics/. The 1000 Genomes phase 3 haplotype data was downloaded from https://mathgen.stats.ox.ac.uk/impute/1000GP_Phase3.html.

**Funding:** CW acknowledges funding from the Wellcome Trust (WT220788) and Medical Research Council (MC_UU_00040/01). J.P. was supported by a Gates Cambridge fellowship (OPP1144). The funders had no role in study design, data collection and analysis, decision to publish, or preparation of the manuscript.

**Competing interests:** CW is a part time employee of GSK and holds shares. GSK had no influence or involvement in this work.

improve the performance of coloc and, when analysing disease risk and gene expression, can clarify which genes the variants that alter disease risk act through.

## Introduction

Genome-wide association studies (GWAS) have uncovered hundreds of thousands of disease associated variants [1,2]. However, more than 90% of these variants lie in non-coding regions of the genome, making elucidating their functional consequence challenging [3,4]. Instead, GWAS associations generally lie in open chromatin regions and are enriched in gene regulatory regions, suggesting they act by modulating gene expression [4–6]. Therefore, a widely used approach to interrogate variant function is assessing whether GWAS and variants associated with gene expression—expression quantitative trait loci (eQTLs)—are caused by the same variants using statistical colocalisation methods [7]. However, most GWAS hits do not have evidence of colocalisation, representing a large 'colocalisation gap' [8].

Today, large amounts of functional annotations and other additional information about genetic variants are available, with initiatives such as The Encyclopedia of DNA Elements (ENCODE) [9] providing genome-wide catalogues of functional annotations. It is well established that GWAS associations are enriched among annotations such as DNase I Hypersensitive sites, implying that functional annotations can provide information about whether variants are causal [4]. More recently, studies have created genome-wide enhancer-gene maps using the activity-by-contact (ABC) score with the intention of prioritising variants that lie in enhancers [10]. Estimates of genomic constraint in non-coding regions have also been generated from large whole genome sequencing datasets, with the idea that highly constrained regions are more likely to have functional consequences [11]. Finally, there are large amounts of publicly available eQTL data that can be used to estimate the most likely positions of eQTLs relative to genes.

Computational methods across statistical genetics have been designed to incorporate this additional variant-specific information to improve performance. For example, in statistical fine-mapping, methods such as PAINTOR [12] incorporate functional annotations in their estimation procedures, while the PolyFun method estimates the prior probability of causality for variants, for use in fine-mapping, based on functional data [13]. A key motivation for using additional data in fine-mapping is that it can 'break ties' between variants in strong linkage disequilibrium, which are otherwise impossible to distinguish. Incorporating functional annotation can greatly concentrates the posterior probabilities on a smaller number of variants. In an analysis of 49 UK Biobank traits, use of PolyFun prior probabilities with the SuSiE fine-mapping method [14], lead to 32% more variant–trait pairs with posterior causal probability >0.95 [13]. Beyond fine-mapping, locus-to-gene models, such as OpenTarget's L2G model, [15] incorporate a range of genetic information, including distance and predicted variant effects. Incorporating distance information was also recently shown to greatly improve peak-to-gene link prediction [16]. However, despite these successes, currently no methods exist for incorporating additional variant-specific information into statistical colocalisation analyses.

Different statistical methods have been developed to assess colocalisation, including Share-Pro, PWCoCo and coloc [7,17,18]. (See [19] for a recent benchmark of Bayesian colocalisation methods). Here, we focus on the widely used coloc method, implemented in the coloc R package, which uses a Bayesian approach to calculate the probability of hypotheses corresponding to different scenarios of shared and distinct causation of associations [7]. Specifically, coloc computes the posterior probability of five hypotheses:

- $H_0$: no association with either trait;
- $H_1$: association with trait 1 only;
- $H_2$: association with trait 2 only;
- $H_3$: association with both traits, distinct casual variants; and,
- $H_4$: association with both traits, shared causal variants.

The probabilities are computed by enumerating all possible configurations of causal variants, assuming there is at most one causal variant for each trait. Each configuration has a prior probability calculated from three variant level prior probabilities:

- $p_1$: the probability each variant is causal for trait 1 only;
- $p_2$: the probability each variant is causal for trait 2 only; and
- $p_{12}$: the probability each variant is causal for both traits.

Previous work has found that the values $p_1 = p_2 = 10^{-4}$, $p_{12} = 5 \times 10^{-6}$ lead to robust inference over a range of scenarios [20]. In addition, coloc was recently extended using the SuSiE fine-mapping algorithm [14] to relax the assumption of a single causal variant in each region [21].

Currently, coloc assumes that the prior probabilities $p_1, p_2$ and $p_{12}$ are uniform across variants in the region of interest. In this paper, inspired by the success of the use of additional variant-specific information in fine-mapping, and with the ultimate goal of narrowing the colocalisation gap, we propose an implementation of variant-specific prior probabilities in coloc. We describe and compare the effectiveness of various sources of information for specifying these prior probabilities using simulated data and pQTL–eQTL colocalisations. Finally, we demonstrate the effect of using variant-specific prior probabilities on eQTL–GWAS colocalisation analyses.

## Results

### Incorporating variant-specific prior probabilities in coloc

We implemented the ability to specify variant-specific priors in the widely used coloc package for statistical colocalisation analysis (Fig 1B). The `coloc.abf()`, `coloc.susie()` and `coloc.bf_bf()` functions in coloc have been updated to include two new arguments: `prior_weights1` and `prior_weights2`, which specify non-negative weights for the probability of a variant being causal for trait 1 and trait 2 respectively. In addition, for consistency we added the ability to use priors weights in the `finemap.abf()` function. Users of coloc can supply any weights they think are informative for the causality of variants *a priori*, including the methods of specifying weights we assess in this manuscript, information about variant level properties in trait relevant contexts, such as single-cell enhancer-gene maps [22], and priors that have been empirically estimated for their trait of interest, for example using Poly-Fun. As coloc does not estimate variant-specific priors, using variant-specific priors is not meaningfully slower than standard coloc (S1 Fig). For mathematical details of the implementation of variant-specific priors see Methods.

### Specification of variant-specific prior probabilities

Various methods are available for specifying variant-specific weights. In this paper we assessed four types of sources for specifying variant-specific weights. Three sources were drawn from public datasets: the output of the PolyFun method [13], 'Gnocchi' score, a measure of non-coding genomic constraint [11] and genome-wide activity-by-contact enhancer-gene link scores [10,23]. For these sources we accessed publicly available data and performed

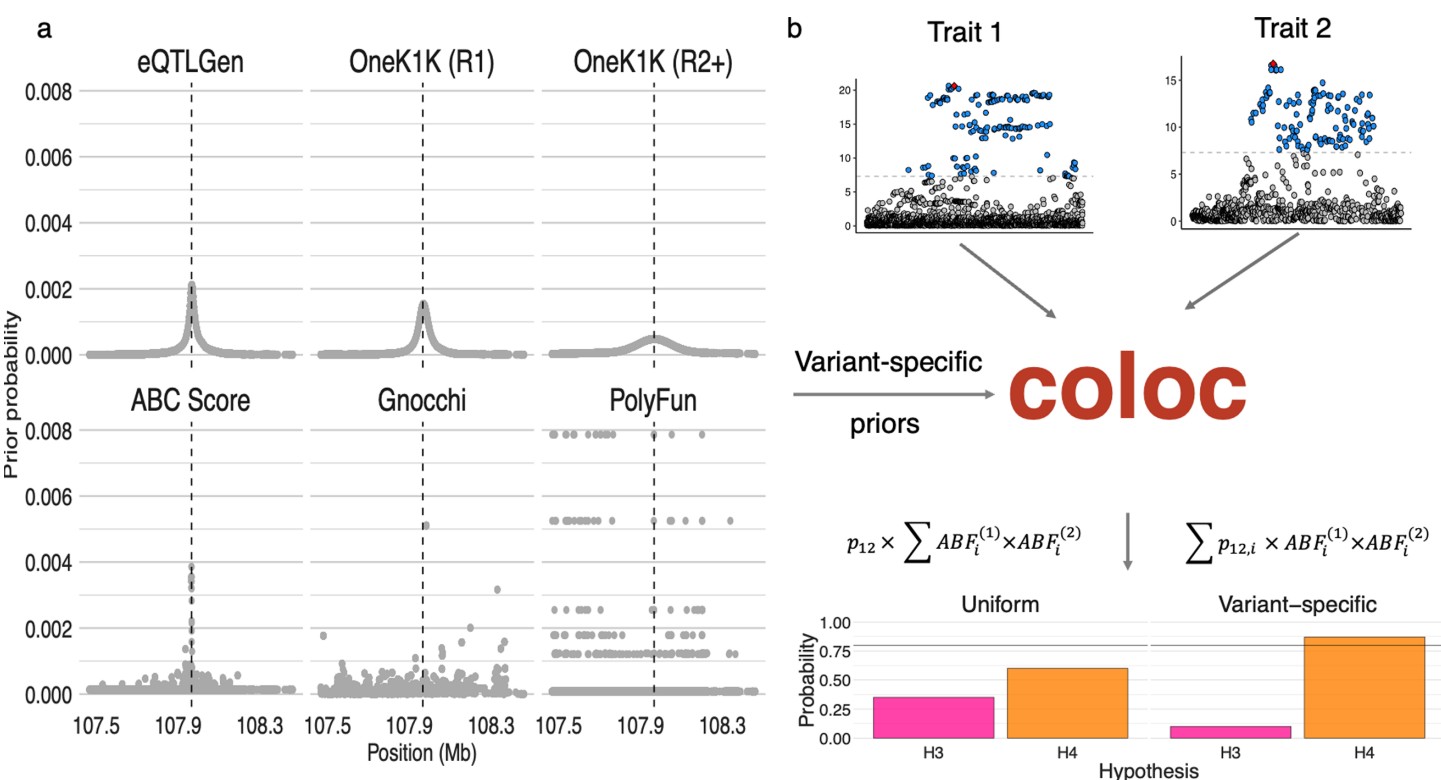

**Fig 1. Example prior probabilities of causality and conceptual figure.** (**a**) Prior probabilities from the six sources of prior probabilities: eQTLGen estimated eQTL-TSS distance density, OneK1K round 1 estimated eQTL-TSS distance density, OneK1K round 2+ estimated eQTL-TSS distance density, genome-wide enhancer-gene link scores calculated using the ABC score, Gnocchi non-coding constraint score and previously calculated PolyFun prior probabilities. The probabilities are displayed in a +/-500kB region around the canonical TSS, shown by the dashed black line, of the *VAV3* genes. (**b**) A conceptual overview of coloc with variant-specific priors. The coloc method takes summary statistic for two traits and test colocalisation, now with the ability to all specify variant specific priors. The formulas shown are proportional to $\Pr(H_4)$ using either variant-specific or uniform priors.

minor processing, mainly to account for SNPs which did not have specific prior values (for further details see Methods). We sought to use empirical eQTL-TSS distance densities as an additional source of prior information. We estimated densities from all significant eQTLs in eQTLGen [24] and both significant round 1 eQTLs and significant round 2-5 conditional eQTLSs OneK1K dataset [25] after finding that the estimated densities were not highly tissues specific (S2 Fig) and that these datasets had more distal eQTLs relative to the TSS compared to other datasets (See Methods for details). Overall, we used 6 sources of variant-specific prior probabilities (Table 1), shown in Fig 1A for a ±500Kb window around the TSS of *VAV3*. The eQTL-TSS density priors place most mass in a small region around the TSS, with the OneK1K round 2+ estimated prior being less concentrated around the TSS. The ABC score-prior is also centred around the TSS, but is less smooth.

## Simulation analysis

First, as a proof of concept, we performed simulations to validate our implementation and highlight the potential benefits of incorporating variant-specific prior information into colocalisation analysis. We simulated association summary statistics for ±500Kb regions around the TSS of three genes with patterns of high LD. High LD regions were chosen as high LD makes determining colocalisation more challenging. Summary statistics were simulated using

**Table 1. Methods for specifying variant-specific prior probabilities.**

| Method | Used for fine-mapping? | Gene-specific? | Dataset | Citation |
|---|---|---|---|---|
| PolyFun | Yes | No | NA | [13] |
| Gnocchi | Yes | No | NA | [11] |
| eQTL-TSS distance | No | Yes | eQTLGen OneK1K round 1 OneK1K round 2-5 | - |
| Activity-by-contact (ABC) score | No | Yes | NA | [10,23] |

simGWAS [26] under both $H_3$ and $H_4$, using the eQTLGen-calculated distance density as the true density of causal variants (See Methods for details). Notably, the simulations recapitulate the tendency of $\Pr(H_4)$ to be close to concentrated close to 0 or 1, for example as in the colocalisation calculated in the OpenTargets Genetics platform. (S4b Fig, see Methods for details). The original version of coloc, which assumes a single causal variant, was run on the simulated data, with and without using the simulation ground-truth distance density as the variant-specific prior.

Overall, using variant-specific priors improved the accuracy of colocalisation. Across all three loci, using variant-specific priors generally increased the computed $\Pr(H_4)$ when the data was simulated under $H_4$ and decreased it when the data was the simulated under $H_3$ (Fig 2A). This finding demonstrates the ability of variant-specific priors to improve colocalisation accuracy, at least when the specified prior matches the true distribution of causal variants. In addition, we assessed how the effect of the variant-specific priors was effected by both the distance of the causal variant to the TSS and the value of $\Pr(H_4)$ when uniform priors were used. We found that, as intended, variants, close the TSS had the largest increase in $\Pr(H_4)$ while distal variants had smaller increases (Fig 2B). We also observed that simulations with moderate values of $\Pr(H_4)$ using uniform priors had the largest change in $\Pr(H_4)$ when variant-specific priors were used (Fig 2C). This observation is partly due to a 'ceiling effect': as $0 \leq \Pr(H_4) \leq 1$, the many values close to 1 cannot be increased by a large amount by the variant-specific priors. The observation also highlights that variant-specific priors may only be able to change whether colocalisation is called at the widely used 0.8 threshold in the minority of cases where $\Pr(H_4)$ is close to 0.8 using uniform priors.

## pQTL-eQTL colocalisation performance comparison

Next, we sought to both evaluate the effectiveness of variant-specific priors in real world data and compare the performance of different sources of prior information. Comparing the performance of the prior-specification methods using simulations is challenging, because it is not clear how to realistically specify the true distribution of causal variant location without using information from prior sources. Furthermore, it is difficult to reliably identify true positives with which to assess colocalisation approaches in real data, as most examples were themselves found using colocalisation analysis, likely biasing the assessment. Instead, we compare the methods on their ability to recover 'ground-truth' pQTL-eQTL colocalisations, inspired by the comparison of colocalisation methods performed in [27]. Methods that detect more same gene pQTL-eQTL colocalisations, while not finding many more different gene pQTL-eQTL colocalisations, should have better performance than methods that do not. We therefore compare the method's ability to recover these 'ground-truth' pQTL-eQTL pairs for protein coding genes. Specifically, we colocalise pQTL for 3,215 proteins detected in blood plasma measured

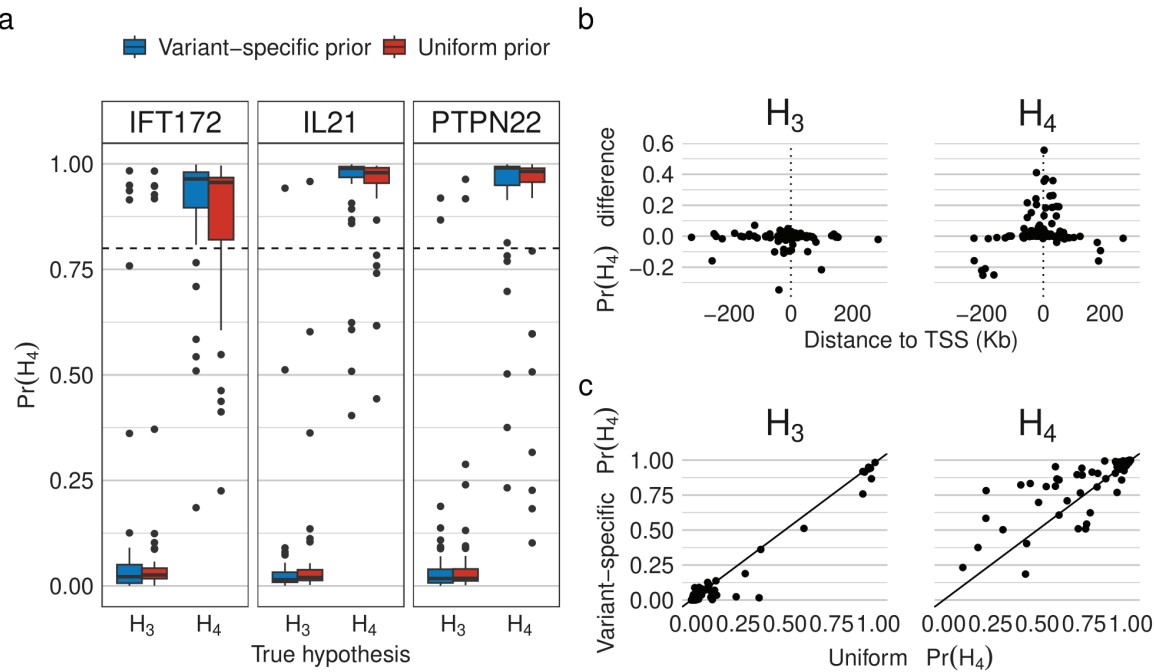

**Fig 2. Simulation analysis.** (**a**) $Pr(H_4)$ values calculated using uniform priors or the simulation ground-truth distribution of causal variants as the variant-specific prior. Simulations are conducted under either coloc hypotheses $H_3$ or $H_4$ in 1MB (+/-500kb) regions around the TSS of the listed genes. (**b**) For $H_3$ and $H_4$ simulations, the $Pr(H_4)$ calculated using variant-specific priors - $Pr(H_4)$ calculated using uniform priors against the position of the simulated causal variant. c) For $H_3$ and $H_4$ simulations, scatter plot of the $Pr(H_4)$ calculated using variant-specific priors against $Pr(H_4)$ calculated using uniform priors.

in 3,301 individuals from the INTERVAL study [28] against eQTL measured in five datasets from the eQTL catalogue (Table A in S1 Text).

We applied all six prior-specification methods to each of the pQTL and eQTL data separately. In addition, for the eQTL TSS densities we applied it to both datasets simultaneously. Colocalisation was performed using both the original version of coloc, which assumes a single causal variant (coloc-single), and the version of coloc which uses SuSiE to relax that assumption (coloc-susie) (See Methods for details). The methods were run on 1Mb windows around the TSSs of protein-coding genes against 5 datasets from eQTL catalogue (Table A in S1 Text, see Methods for more details). We applied the eQTL-TSS distance density priors centred at the TSS of the eQTL gene when applied to the eQTL dataset and centred at the TSS of gene associated with the protein applied to the pQTL dataset. We summarised the results of the analysis in two ways. First, we calculated the recall and precision of methods at the $Pr(H_4) > 0.8$ threshold for colocalisation, calling colocalisation if the pQTL colocalised across any of the eQTL datasets (Fig 3A and 3C). (For a discussion of the use of the $Pr(H_4) > 0.8$) threshold for calling colocalisation see Methods.) Second, we used a ROC-style analysis to assess the performance of including the prior information over a range of $Pr(H_4)$ thresholds (Fig 3B and 3D). (See Methods for details).

There were three main findings from this analysis. First, the variant-specific priors had a positive but modest improvement in performance for both coloc-single and coloc-susie. At the 0.8 threshold, the maximum improvements in recall from incorporating prior information was 0.04 and the maximum improvement in precision was 0.03. Prior information had a larger effect on performance when $Pr(H_4)$ threshold was lower, corresponding to higher true

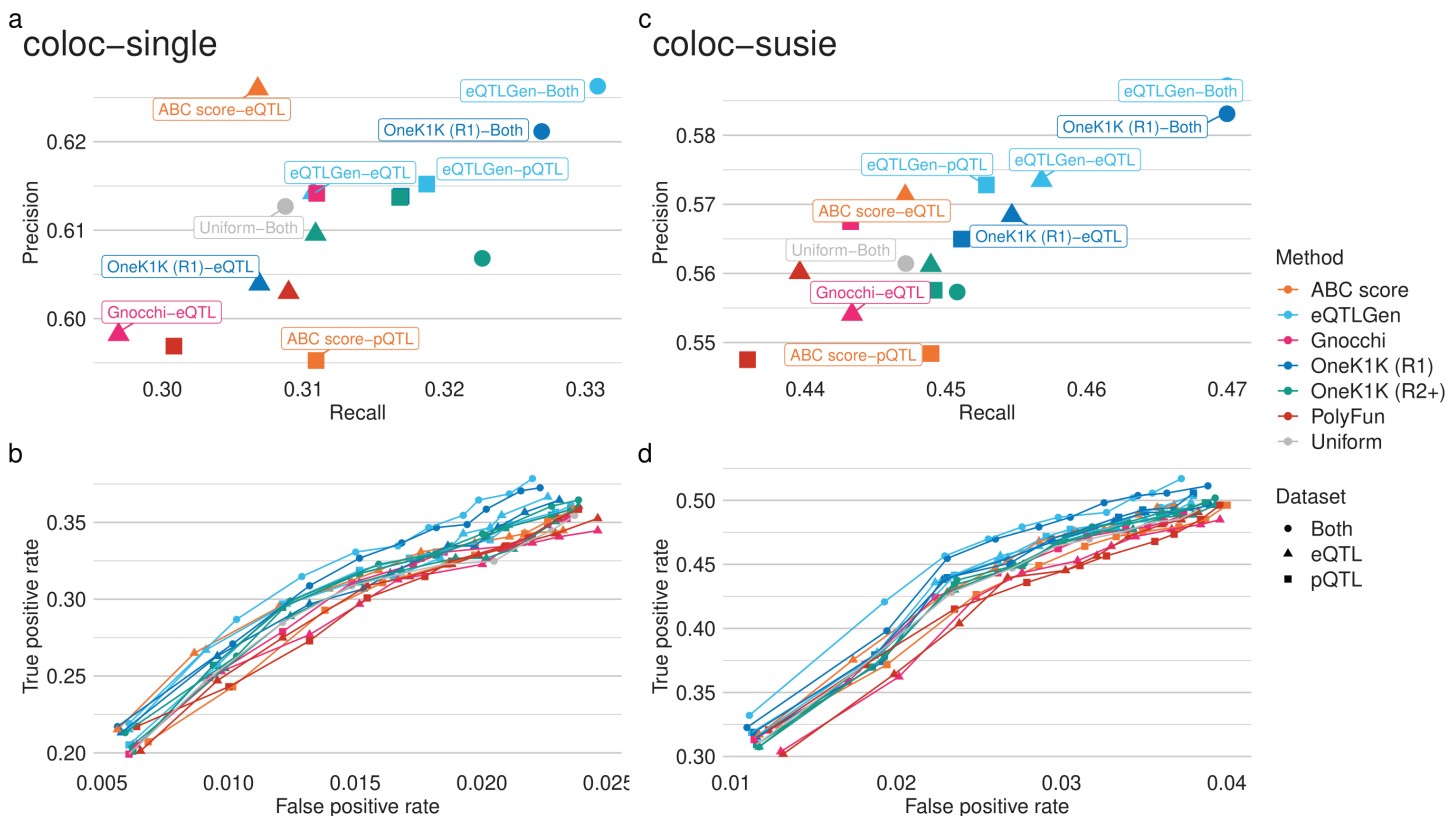

**Fig 3. pQTL-eQTL colocalisation performance assessment for coloc with a single causal variant assumption.** (**a**) Recall and precision of colocalisation at the $\Pr(H_4) > 0.8$ significance threshold based on ground-truth pQTL-eQTL colocalisations. All 6 different sources of prior information were applied to either the eQTL or pQTL data, with the 3 eQTL-TSS distance density priors additionally applied to both datasets. (**b**) A ROC-style analysis of the ground-truth pQTL-eQTL data using the $\Pr(H_4)$ thresholds: 0.5, 0.55, ..., 0.95. In both panels coloc assuming a single causal variant (coloc-single) is used. (**c**) As in (a) but using coloc with SuSiE (coloc-susie). (**d**) As in (b) but using coloc with SuSiE (coloc-susie).

and false positive rates. At the 0.8 threshold, all sources of prior information improved recall but at the cost of precision for the ABC score prior applied to the pQTL dataset, OneK1K Round 2 and Gnocchi priors applied to the eQTL dataset. Second, when applied to one dataset only, the best performing priors at the 0.8 threshold were the eQTLGen-estimated density prior applied to the eQTL and pQTL datasets, the OneK1K R1-estimated density prior applied to the pQTL dataset and the ABC score prior applied to the eQTL dataset. The worst performing priors were the PolyFun prior applied to the eQTL dataset, the OneK1K round 2-estimated prior applied to the eQTL dataset and the ABC score prior applied to the pQTL dataset. Across thresholds, the eQTL-estimated priors shows the best performance. Finally, we observed much better performance when the eQTL-estimated priors were applied to both pQTL and eQTL datasets simultaneously. This good performance makes sense as applying the priors to both datasets strongly encodes our knowledge that the pQTL and eQTL signals for the same gene/protein should be close. Furthermore, comparing results for coloc-single and coloc-susie, the different priors had similar relative effects on performance and the overall ordering of priors was the same for the two methods. However, coloc-susie had substantially higher recall but a slightly decreased precision (Fig 3C and 3D).

To better understand the modest performance improvement of the priors in this comparison we assessed their overall impact on the posterior probability of colocalisation. We

observed that all the priors had only a small effect on the calculated value of $\Pr(H_4)$ (absolute value of change in $\Pr(H_4)$ less than 0.01) in over 80% of loci. (S5A Fig). Using the variant-specific priors with coloc with SuSiE had a smaller effect on $\Pr(H_4)$ across methods compared to coloc with the single causal variant assumption, particularly when applied to the eQTL datasets. We ascribe this difference to the higher maximum log Bayes factors calculated by SuSiE compared to calculated by the coloc-single method, suggesting that there is more information in the likelihood for coloc-susie (S5B Fig). Similarly, we observed that the magnitude of log Bayes factors were related to the size of the effect of variant-specific priors, with large effects only occurring when the maximum was small (S5C Fig). Focusing on the impact of the eQTLGen density prior, we observed that the values of $\Pr(H_4)$ with uniform and variant specific priors, were highly correlated across loci, with only a minority of loci displaying large enough changes to alter conclusions at the 0.8 threshold (S5B Fig). Threshold effects explain some of the lack of changes, but they do not explain the absence of loci where the variant-specific prior substantially reduces $\Pr(H_4)$ (S5C Fig). We hypothesised that the lack of changes, unlike in finemapping, could be due to colocalisation posterior probabilities being more concentrated towards 1 than probabilities produced by fine-mapping. However, comparing colocalisation posterior probabilities and SuSiE fine-mapping causal probabilities in the OpenTargets catalogue, indicated that both types of probabilities are highly concentrated close to 1 (S4 Fig).

Overall, this analysis showed that incorporating prior information improves colocalisation performance in real colocalisations but the magnitude of the improvement is modest. The best performing source of prior information were the empirical eQTL-TSS distance densities.

## GWAS-eQTL colocalisations analysis

Finally, we assessed the impact of variant-specific priors on GWAS-eQTL colocalisation in order to understand how colocalisation results might change with the use of distance priors on eQTL studies. We performed colocalisation between ten FinnGen (Release 10) GWAS and five eQTL datasets for disease-relevant tissues for each trait drawn from the eQTL catalogue (Table B in S1 Text, see Methods for details). The GWAS traits were chosen to represent a wide variety of traits, including two quantitative traits (height and weight) and three traits with lower case numbers (IBD, type 1 diabetes and rheumatoid arthritis). The colocalisation results using a uniform prior were compared to those using the eQTLGen and OneK1K round 1-estimated eQTL–TSS distance priors, which had the best performance in the pQTL–eQTL comparison.

Across trait-dataset pairs the variant-specific priors had a moderate impact on changing whether a variant-gene pair was significant (Fig 4A), with at most 14.1% of colocalisations with some evidence of colocalisations ($\Pr(H_4) > 0.5$)) changing significance at the 0.8 threshold. For coloc with the single causal variant assumption, height, hypertension and IBD had the highest proportion of colocalisations change significance while for coloc-susie weight and Type 1 diabetes had the highest proportion. Notably, across priors and colocalisation methods no colocalisations for rheumatoid arthritis changed significance. Overall, less colocalisations changed when the priors were used with coloc with SuSiE compared to coloc with the single variant assumption, while the different eQTL-TSS distance priors had very consistent effects. Examining the effect of priors on $\Pr(H_4)$ showed that the for most loci the priors had little effect (Fig 4B), particularly for coloc with SuSiE (Fig 4C). Colocalisations that changed significance for either coloc-single (S1 Table) or coloc-susie (S2 Table) are recorded in the supplementary materials.

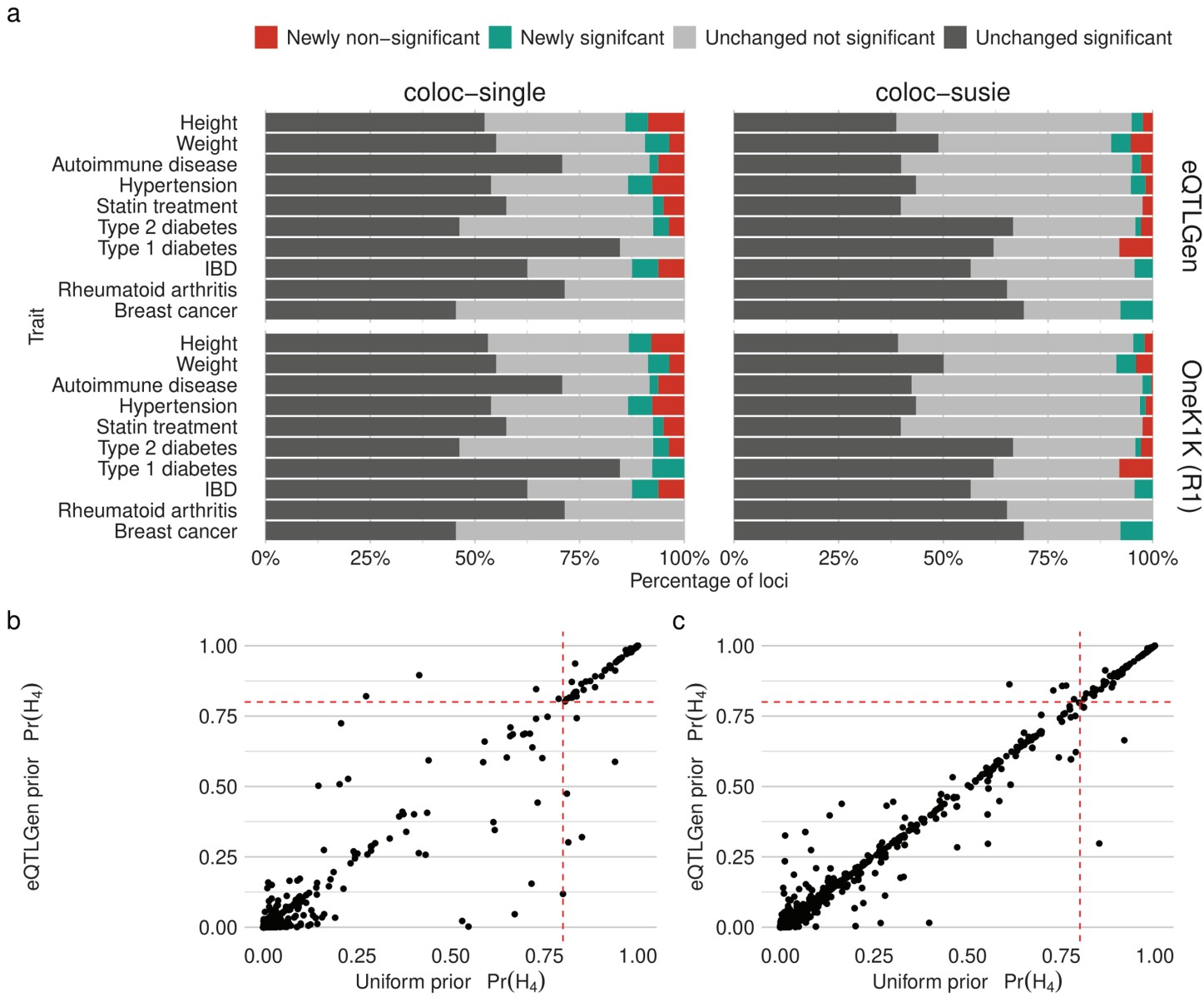

**Fig 4. GWAS-eQTL colocalisation analysis.** (**a**) Number of loci with some evidence of colocalisation ($\Pr(H_4) > 0.5$) using a uniform prior coloured by the effect of a variant-specific priors on colocalisation significance threshold of 0.8. Results are for the eQTLGen and OneK1K round 1-estimated priors and coloc using SuSiE (coloc-susie) and coloc with the single variant assumption (coloc-single). Unlike in the pQTL-eQTL analysis, here colocalisations with each credible set are counted as separate colocalisations. (**b**) Scatter plot of $\Pr(H_4)$ computed using uniform and eQTLGen eQTL-TSS distance density priors for coloc with the single causal variant assumption applied to Autoimmune disease trait colocalisatons. (**c**) As in (**b**) but using coloc with SuSiE.

Given the strong reported performance of PolyFun for fine-mapping we also investigated whether PolyFun had a larger impact on colocalisation significance when its prior probabilities were estimated in a trait-specific manner. We estimated trait-specific prior probabilities for three traits in the UK Biobank—Type 2 diabetes, Hypertension and Hypothyroidism (see Methods for details). We performed this analysis only for UK Biobank traits as the PolyFun prior estimation is very computationally demanding and requires additional functional annotation and LD information which is not readily accessible outside the UK Biobank. First,

visualising the priors in the same region as Fig 1 (S6 Fig), showed that the trait-specific priors had a broadly similar form to the precomputed priors, although with more variant at smaller values. Next, we assessed the impact of the precomputed and traits-specific priors on credible set size (S7A Fig). As expected, the PolyFun priors were able to decrease the credible set size compared to a uniform variant prior. However, we observed that the precomputed priors generally had a larger impact than the trait-specific priors. Finally, we performed colocalisation analysis between the three traits and five eQTL datasets in disease relevant tissues (Table C in S1 Text, see Methods for details). In this analysis we again found that the trait-specific priors had less impact than the precomputed prior on colocalisation significance (S7B Fig). Taken together, these observations suggest that trait-specific PolyFun priors are unlikely to outperform the precomputed priors, cautioning against there use, especially given the large computational cost of estimating the trait-specific priors.

One GWAS signal for which using variant-specific priors changed colocalisation conclusions was a FinnGen automimmune disease trait signal near the genes *PSMB7* and *NEK6*. Using coloc-single, this signal strongly colocalised with eQTL associations for both *PSMB7* and *NEK6* in GTEx thyroid tissue (Fig 5D). Extended LD means the GWAS signal is spread across both genes although the peak along this "plateau" is nearer the NEK6 TSS than the PSMB7 TSS. In the eQTL data, the peak of each gene signal is close to their respective TSSs. (Fig 5A and 5B and 5C). When the eQTLGen prior, centred at the *NEK6* TSS, is applied to the GWAS-*NEK6* colocalisation it remains significant (Pr($H4$) = 0.98), but when the prior, centred at the *PSMB7* TSS, is applied to the GWAS-*PSMB7* colocalisation its Pr($H_4$) decreases to 0.42. Therefore, incorporating prior information at this locus changes the conclusion of colocalisation to strongly prefer *NEK6* as the causal gene. This example highlights the ability of variant-specific priors to resolve multiple colocalisations at a locus in a natural, distance-dependent way.

## Discussion

In this paper we described and validated an implementation of variant-specific prior probabilities in the widely-used coloc method for statistical colocalisation analysis. We assessed whether universally applicable sources of prior information, such as the estimated distance of eQTLs to the TSS could be used to improve colocalisation performance. Our simulations and pQTL-eQTL colocalisation analysis demonstrated that incorporating prior information can improve the accuracy of colocalisation, but the improvements were modest, with 2-3% improvements in recall and precision in comparisons based on ground truth pQTL–eQTL colocalisations. In this context, pQTL–eQTL-based comparisons are very useful to compare the effect of varying priors for coloc, and there is no equivalent ground truth dataset in fine-mapping. However, we note that our analysis is likely to overestimate false negatives, because the proteins in plasma originate from a wide array of tissues and our eQTL datasets cover only a subset of these. Overall, the best performing source of prior information were the estimated eQTL-TSS distance densities. In GWAS-eQTL colocalisations, using the eQTL-TSS density as source of variant specific prior information changed colocalisation significance in up to 13.5% of all loci with some evidence of colocalisation (Pr($H_4$) > 0.5). The eQTL-TSS density priors could sensibly distinguish between multiple gene with significant colocalisations, for example at the *NEK6/PSMB7* locus. Furthermore, as variant-specific prior information must be supplied to coloc, improvements in performance gains come at essentially no computational cost. We also anticipate that users may observe variant-specific priors having a larger effect on colocalisation if they can supply priors that contain substantial information about the genetic architecture of the trait, such as single-cell enhancer-gene maps in trait relevant tissues [22].

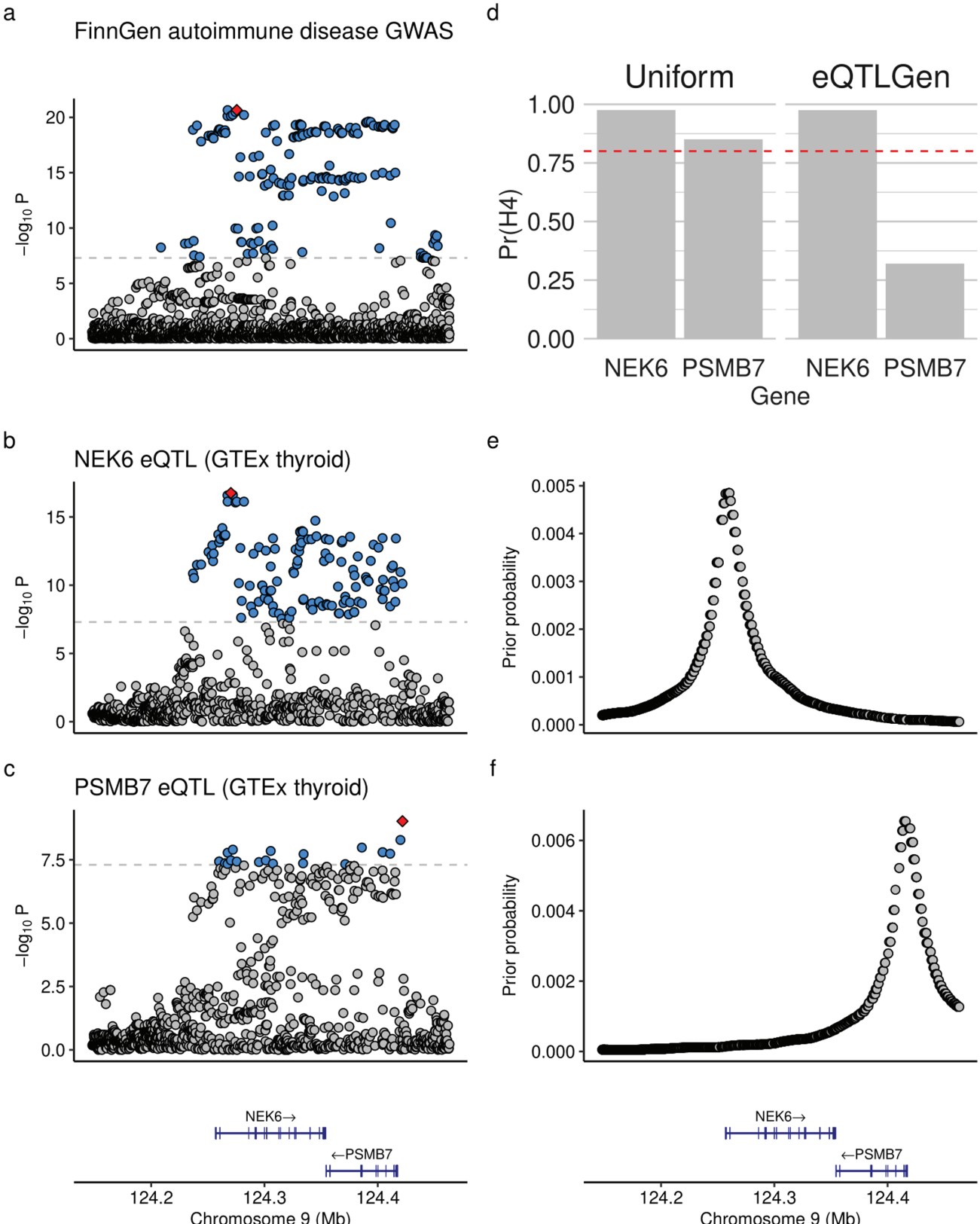

**Fig 5. Colocalisation at the *NEK6-PSMB7* locus.** (**a**) Manhattan plot of the FinnGen v10 'Autoimmune disease' trait GWAS in a ±500Kb region around the TSS of *NEK6*. (**b**) Manhattan plot of the *NEK6* GTEx thyroid eQTL results in the same region. (**c**) Manhattan plot of the *PSMB7* GTEx thyroid eQTL

results in the same region including gene track information showing *NEK6* and *PSMB7*. (**d**) $\Pr(H_4)$ returned by coloc for colocalisation between the 'Autoimmune disease GWAS' and the *NEK6* and *PSMB7* eQTL results using both uniform priors and eQTLGen-estimated eQTL-TSS distance priors for each genes. (**e**) The eQTLGen-estimated eQTL-TSS distance prior probabilities for *NEK6*. (**f**) The eQTLGen-estimated eQTL-TSS distance prior probabilities for *NEK6* with the same gene track as in (c).

Finally, we note that using variant-specific prior probabilities is not expected to replace the need to perform colocalisation using a wider range of tissue- and context-specific eQTLs and more diverse molecular trait QTLs.

In contrast to previous reports in fine-mapping [13], all variant-specific priors did not alter conclusions in a large number of cases . We speculate this lack of effect is due to fine-mapping being more sensitive to priors probabilities than colocalisation. For example, a fine mapping prior might only need to up-weight one variant to change a credible set of size two to size one and clarify the causal variant. However, to affect the probability of colocalisation, which is computed by summing across variants, we likely need to change the posterior probabilities for the majority of variants that provide evidence of colocalisation. This interpretation is supported by the greater impact of the eQTL-TSS distance density priors, which consistently alter prior probabilities for a large proportion of variants at a locus, compared to other sources of prior information. However, we note our findings are consistent with the reported effect of using the Gnocchi score to specify variant-specific priors for fine-mapping where only 0.3% of variants were newly identified as likely causal (PIP >0.8) [11]. In addition, in our comparisons the variant-specific priors, other than the eQTL-TSS density distance priors, did not consistently improve eQTL-pQTL colocalisation analysis. This lack of improvement may be because the priors do not capture of position of causal eQTL variants, for example because of systematic differences between eQTLs and GWAS hits [8]. Our comparison highlights the lack of systematic comparison of different strategies for setting prior probabilities in statistical fine-mapping.

In this paper we incorporated additional information at the variant level in colocalisation analysis by altering prior probabilities. The limited impact of this approach suggests that other methods of incorporating additional information may be preferable. One approach would be to introduce additional information at the GWAS analysis stage, capturing that information in updated Bayes factors that could be used by coloc. Another approach is to incorporate additional information after colocalisation has been performed, as in locus-to-gene models such as OpenTarget's L2G model [15], which use colocalisation probability as a covariate along with information such as distance and predicted variant effect.

Furthermore, this paper has a two main limitations. First, the pQTL-eQTL results of colocalisation comparison may not perfectly generalise to the GWAS-eQTL setting, due to the differences in genetic architecture between e/pQTL and GWAS hits [8]. Second, all QTL datasets in this study are drawn from primarily European-ancestry cohorts, reflecting a general bias in currently available QTL data. The use of data from non-European-ancestry individuals may improve the performance of colocalisation analysis, especially for GWAS performed in individuals of diverse ancestry. We note that coloc does not require both datasets to have the same LD structure so it can be used to perform cross-ancestry colocalisation analyses.

The ability to use variant-specific priors has been implemented in version 6.0.0 of coloc. Based on the analysis conducted in this paper, we recommend that eQTL-TSS distance densities priors be considered for use in all GWAS-eQTL colocalisations. In our comparisons, other sources of prior information did not have a large enough impact to justify their use. However, our results do not preclude the use of additional bespoke sources of information, such as readouts from disease relevant cell-types, which we did not explore due to a lack of ground-truth

benchmark. Any future sources of prior information that are demonstrated to improve fine-mapping results could be tried in colocalisation analysis. We expect that variant-specific priors will also be useful to better understand colocalisations of interest, such as colocalisations that show borderline significance. Practically, we advise that variant-specific priors should only be applied to one trait at a time, unless there is strong independent prior information for both traits. (See the Supplementary Text for more discussion of applying priors to both traits.) Finally, we emphasise that any bespoke variant-specific priors used in an analysis should be transparently reported and explained, and their use justified in detail.

## Methods

### Variant-specific priors in colocalisation

The `coloc.abf()`, `coloc.susie()` and `coloc.bf_bf()` functions in coloc have been updated to include two new arguments: `prior_weights1` and `prior_weights2`, which specify non-negative weights for the probability of a variant being causal for trait 1 and trait 2 respectively. These weights are used to calculated variant-specific prior probabilities, according to the formula

$$p_{k,i} = Q \times p_k \times \frac{w_{k,i}}{\sum_{i=1}^{Q} w_{k,i}}, \; k \in \{1,2\}, \; i \in \{1,\dots,Q\}, \tag{1}$$

where $w_{k,i}$ is the provided weight for SNP $i$ and trait $k$, $Q$ is the number of variants analysed, and $p_{k,i}$ is the prior probability of SNP $i$ for trait $k$. The variant-specific prior probabilities of being causal for each trait are then used to calculate the variant-specific prior probability of causality for both traits

$$p_{12,i} = \frac{p_{12}}{p_1 p_2} \times p_{1,i} \times p_{2,i} \tag{2}$$

where $p_{12}$, $p_1$ and $p_2$ are as described above. This calculation of the prior probabilities is designed to ensure that the variant-specific prior probabilities satisfy two conditions: that they are proportional to $w_{k,i}$, and that they encode the same hypothesis prior probabilities as uniform priors. To see that this is the case, note that with uniform priors,

$$\Pr(H_k) \approx Q \times p_k, \; k \in \{1,2\} \tag{3}$$

while, with variant specific priors,

$$\Pr(H_k) = \sum_{i=1}^{Q} p_{k,i} = \sum_{i=1}^{Q} Q \times p_k \times \frac{w_{k,i}}{\sum_{i=1}^{Q} w_{k,i}} = Q \times p_k, \; k \in \{1,2\} \tag{4}$$

Therefore, the variant-specific priors incorporate information from the supplied variant-specific weights while not affecting the overall prior probability of the colocalisation hypotheses. When one of the priors, $p_1$ say, is variant-specific then we have

$$\Pr(H_4) = \sum_{i=1}^{Q} p_{12,i} = \sum_{i=1}^{Q} \frac{p_{12}}{p_1 p_2} \times p_{1,i} \times p_2 = \sum_{i=1}^{Q} \frac{p_{12}}{p_1} \times Q \times p_1 \times \frac{w_{1,i}}{\sum_{i=1}^{Q} w_{1,i}} = Q \times p_{12}, \tag{5}$$

so the overall prior weight on $H_4$ is the same as with uniform priors. See the Supplementary Text for an introduction to the coloc method and further details about the case where both priors are variable.

### Variant-specfic priors speed benchmark

We compared the speed of coloc with and without variant-specific priors using simulated data included in the coloc package (with 500 and 2000 simulated variants) and randomly generated prior weights. Execution speed was measured using the R package bench.

### Prior probability sources

**Overview.** We use four sources of prior probabilities in this paper. First, we use prior probabilities of causality computed by the PolyFun method [13] applied to 19 million imputed UK Biobank SNPs with MAF >0.1%, based on a meta-analysis of 15 UK Biobank traits. Poly-Fun is a method designed to compute prior probabilities in proportion to predicted per-SNP heritabilities given functional annotations. In an additional analysis of data from the UK Biobank we also compute trait-specific PolyFun priors. Second, we used the 'Gnocchi' score, a measure of non-coding genomic constraint estimated from 76,156 human genomes [11]. The idea of the score is that regions that show high constraint are more likely to have functional consequences; use of the Gnocchi score in fine-mapping of the UK Biobank association data increased posterior inclusion probabilities for a subset of variant-trait pairs [11]. Third, we used genome-wide enhancer-gene link scores calculated using the activity-by-contact (ABC) method [10,23]. The ABC score measures the strength of enhancer-gene connections based on the activity of the enhancer and the contact between the gene and the enhancer. The contact is measured using genome-wide chromatin conformation capture techniques (Hi-C), that measure whether two DNA fragments physically associated in 3D space [29]. Finally, we used the empirical density of the distance of eQTLs to their associated TSSs we calculated from public datasets.

**ABC score.** The ABC score for a gene $G$ contributed by element $E$ is,

$$\text{ABC score}_{E,G} = \frac{A_E \times C_{E,G}}{\sum_e A_e \times C_{e,G}}, \tag{6}$$

where $A_E$ is the activity, its strength as an enhancer, of element $E$ and $C_{E,G}$ is the Hi-C measured contact between element $E$ and gene $G$. The sum in the denominator is taken over all elements within 5 Mb of $G$ [23].

We use ABC data collected for 131 biosamples generated by [10], which includes details of how activity and contact are measured. The dataset only includes gene-element connections with an ABC score $\geq 0.015$. To convert the dataset into a vector of prior probabilities we filter to the gene of interest, take the median over all biosamples and set the value for a SNP to the value of the enhancer it lies in. If the SNP does not have a listed score we set the value to 0.0075, half the minimum score.

**PolyFun.** PolyFun calculates prior causal probabilities that are proportional to the 'per-SNP heritability' estimates, $\text{var}(\beta_i \mid \boldsymbol{a}_i)$, that is it sets,

$$\Pr(\beta_i \neq 0 \mid \boldsymbol{a}_i) \propto \text{var}(\beta_i \mid \boldsymbol{a}_i), \tag{7}$$

where $\beta_i$ is the effect size for SNP $i$ and $\boldsymbol{a_i}$ is the vector of functional annotations for SNP $i$. This relationship can be derived under a point-normal prior model for $\beta_i$ and assuming that the causal variance is independent of $\boldsymbol{a_i}$; that is that $\text{var}(\beta_i \mid \beta_i \neq 0, \boldsymbol{a_i}) = \text{var}(\beta_i \mid \beta_i \neq 0)$ [30].

*Precomputed*

To use precomputed PolyFun weights we downloaded the weights pre-computed using 9 million imputed UK Biobank SNPs with MAF> 0.1%, based on a meta-analysis of 15 UK Biobank traits. We re-implemented the processing performed in `polyfun/extract_snpvar.py` to link the weights to data. SNPs that did not have a variance entry were assigned the minimum entry present in the data.

*Trait-specific*

To compute trait-specific prior PolyFun prior probabilities we followed the PolyFun approach 3 instructions in the wiki of the PolyFun GitHub repository. Briefly this involved:

1. Munging summary statistics into PolyFun friendly format
2. Running PolyFun with L2-regularized S-LDSC using baseline annotations produced by the PolyFun authors [31]
3. Computing LD scores for each SNP bin using LD calculated in the UK Biobank provided for use in PolyFun.
4. Re-estimating per-SNP heritabilities via S-LDSC.

After computation of the PolyFun priors probabilities we used liftover to lift them from hg19 to hg39.

Running PolyFun to compute trait-specific prior probabilities was very computationally intensive. Running PolyFun with L2-regularized S-LDSC required over 100GB of RAM per chromosome-specific job, while computing LD scores for each SNP bin required genome-wide LD information, which was approximately 1TB on disk.

**Gnocchi.** Gnocchi is a signed score that quantifies the depletion of genomic variation at 1Kb scale [11]. Precisely, the score is defined as,

$$\text{Gnocchi} = \begin{cases} \sqrt{\chi^2} \text{ if } O < E, \\ -\sqrt{\chi^2} \text{ if } O \geq E \end{cases} \tag{8}$$

where $\chi^2 = (O - E)^2/E$ and $E$ and $O$ are the observed and expected values of the variation, respectively. The expected amount of variation is calculated based on estimated probabilities of mutations in trinucleotide contexts and adjusted for DNA methylation and genomic features. These calculations used data on 76,156 human genomes from gnomAD.

To assign variants to Gnocchi score we used the following approach. If a variant lay in a region with a Gnocchi score it was assigned that score. Otherwise it was assigned the score of the closest region with a Gnocchi score. The assigned scores were then converted to non-negative weights by a softmax transformation.

**eQTL-TSS distance density.** To compute an empirical eQTL-TSS distance density we used several publicly available datasets: the datasets in the eQTL Catalogue [32], GTEx v8 (accessed through the eQTL catalogue) [33], eQTLGen [24] and the OneK1K dataset [25].

To estimate the eQTL-TSS distance density from summary statistics, we used the following strategy. First, we filtered the eQTL summary statistics to only include genome-wide significant ($p < 5 \times 10^{-8}$) eQTLs, and only the most significant eQTL for each gene. (This approach was inspired by [34]) Information about the TSS and strand of each gene was downloaded from Ensembl with the `biomaRt` R package [35]. If a gene had multiple listed TSSs

the median of their locations was used as the TSS. For each eQTL with position $p$ say, the distance, $d$ to the TSS, with position $t$ say, was calculated as

$$d = \begin{cases} p - t, & \text{if + strand gene} \\ t - p, & \text{if – strand gene.} \end{cases} \qquad (9)$$

To compute the density from the calculated distances we used a non-parametric approach to ensure maximum fidelity. We used the `density()` function in R, with non-default parameters `bw = "SJ", cut = 0, adjust = 8`. The parameter `bw = "SJ"` specifies an alternate bandwidth selection algorithm that is recommended in the R documentation. The parameter `cut` specifies how far beyond the extremes of the data the density should become approximately 0. In practice, we found that values of cut >0 produced a small number of points with much smaller values compared to the rest of the estimated density. The parameter `adjust` specifies a multiplicative scaling factor for the bandwidth. We set `adjust = 8` to heuristically account for uncertainty in the location of the TSS. The estimated density is represented computationally as 512 (distance, density value) pairs. The densities were computed in a $\pm 500$kb window around the TSS.

To determine which datasets to use the estimated densities of we assessed the distribution of distances from the TSS across all the datasets (**??**). This analysis showed relatively little variation between tissues in GTEx or the eQTL catalogue, suggesting that tissue-specific densities are not needed. It also highlighted that eQTLs in the datasets eQTLGen and OneK1K were generally more distal from the TSS. Investigating the OneK1K dataset further, we found that eQTLs that were specific to a cell type (**??**c) and identified in subsequent rounds of conditional mapping (**??**a) were more distal [36] while distance was consistent across cell types (**??**b) except for a few rare cell-types. Therefore, based on evidence that GWAS variants lie further from gene TSSs than eQTL [8] and empirical evidence conditional estimation can lead to more colocalisations [36] we selected the eQTLGen, cell-type specific OneK1K Round 1 and OneK1K (Round 2-5) as the being the most appropriate datasets/data subsets to estimate prior information from.

## Colocalisation significance threshold

In this paper, we use the threshold of $\Pr(H_4) > 0.8$ as the standard threshold for calling a colocalisation as significant. The threshold was first used in the original paper describing coloc, where colocalisations with > 80% (i.e. >0.8) $\Pr(H_4) > 0.8$ are referred to as 'positive colocalisation results' [7]. Today, the threshold is widely used to call colocalisations as significant in applied paper that use coloc (see, for example, [37–39]). Additionally, an empirical attempt to evaluate this threshold concluded that both the coloc $H_4 > 0.8$ and GWAS $p < 5 \times 10^{-8}$ thresholds achieved similar performance in predicting validation in larger GWAS from results of a smaller GWAS [40]. Finally, other colocalisation methods such as SharePro also use the 0.8 threshold [18].

## Simulation analysis

We simulated GWAS summary data under a single-causal variant assumption, simulating data under both $H_3$ and $H_4$. We used haplotypes for EUR samples in the 1000 Genomes phase 3 data [41], phased by IMPUTE2 [42], (https://mathgen.stats.ox.ac.uk/impute/1000GP_Phase3.html)

We used the `simGWAS` R package method to simulate GWAS summary statistics with the LD and MAF calculated from the haplotypes [26]. We simulated case-control data with 2,000 cases and 2,000 controls. The effect size of the causal variant was simulated as the maximum of 100 $N(0,0.0025)$ random variables. In each call to `simGWAS()` the simulation was repeated 2000 times and only the first simulation that had a minimum p-values less than $5 \times 10^{-6}$ was used, matching our expectation that colocalisation is only performed when there is at least a moderate signal of association. Specifically we simulated data for $\pm$500Kb windows around the TSS of three genes: *IL21*, *PTPN22*, *IFT172*. To simulate variant-specific priors we used the following algorithm. First, we sample the causal variant for the first trait from the eQTL-Gen density variant-specific prior. Second, sample the causal variant for the second trait, depending on the hypothesis:

- $H_3$: sample the causal variant uniformly from all variants other than the causal variant for trait 1 or,
- $H_4$: set the casual variant to the causal variant for trait 1.

## pQTL-eQTL colocalisation analysis

We used the pQTL dataset produced by mapping QTLs for 3,215 measured proteins in blood plasma from 3,301 individuals who were part of the INTERVAL study [28]. We colocalised the dataset against 5 eQTL datasets from the eQTL Catalogue (Table A in S1 Text) chosen to maximise the number of colocalisations with plasma pQTLs. Summary statistics and fine-mapping credible set and log Bayes factor (LBF) files were downloaded from the eQTL Catalogue.

Using `coloc.abf()`, we performed colocalisation in a 1MB window around all protein-coding genes. Colocalisation was only performed if the region contained at least one pQTL with p-value $< 5 \times 10^{-8}$ and at least one eQTL with p-value $< 5 \times 10^{-6}$. The `coloc.abf()` function was run prior probabilities: $p_1 = p_2 = 10^{-4}, p_{12} = 5^{-6}$.

Using `coloc.susie()`, we performed colocalisation in a 1MB window around all protein-coding genes. Colocalisation was only performed for credible sets in both pQTL and eQTL datasets. The `coloc.bf_bf()` function was run with default prior probabilities ($p_1 = p_2 = 10^{-4}, p_{12} = 5 \times 10^{-6}$)

For both approaches a colocalisation was defined as significant if the maximum colocalisation across eQTL datasets was $\Pr(H_4) > 0.8$. In the coloc with SuSiE analysis the maximum colocalisation was also taken across credible set pairs. To assess the performance of the methods we calculated various measures of classifier performance. To calculate these metrics we defined true positive, false positives, true negatives and false negatives in the following way,

- True positive: significant colocalisation between pQTL and eQTL for the same gene.
- False positive: significant colocalisation between pQTL and eQTL for different genes.
- False negative: no significant colocalisations but pQTL and eQTL genes match.
- True negative: no significant colocalisation and pQTL and eQTL genes do not match.

These metrics are only calculated for regions where colocalisation is performed i.e. where at least one pQTL with p-value $< 5 \times 10^{-8}$ and at least one eQTL with p-value $< 5 \times 10^{-6}$.

First, following the analysis in [27] we calculated the recall and precision of each prior method. These values were

$$\text{Recall} = \frac{\text{TP}}{\text{TP} + \text{FN}}, \quad \text{Precision} = \frac{\text{TP}}{\text{TP} + \text{FP}} \tag{10}$$

where TP is the number of true positives, FN is the number of false negatives and FP is the number of false positives. Second, we compute a receiver operator curve (ROC) for each method, for a series of colocalisation significance thresholds $(0.5, 0.55, \ldots, 0.95)$ for calculating the true positive rate (TPR) and false positive rate (FPR). These metrics are calculated as

$$\text{TPR} = \frac{\text{TP}}{\text{TP} + \text{FN}}, \quad \text{FPR} = \frac{\text{FP}}{\text{FP} + \text{TN}} \tag{11}$$

where FP is the number of false positives, FN is the number of false negatives and FN is the number of false negatives.

### FinnGen GWAS-eQTL colocalisation analysis

Colocalisation was performed for ten FinnGen traits against 5 eQTL datasets in disease-relevant tissues per trait (Table B in S1 Text). We used GWAS summary statistics and fine-mapping credible set and log Bayes factor (LBF) files from FinnGen Release 10. In this analysis colocalisations with a signal captured by a single credible set are counted separately, unlike in the pQTL–eQTL analysis where the maximum is taken over colocalisations with all credible sets in a locus.

Using `coloc.abf()`, we performed colocalisation in a 1MB window around all protein-coding genes. Colocalisation was only performed if the region contained at least one genome-wide significant GWAS SNP and at least eQTL with p-value less than $5 \times 10^{-6}$. The `coloc.abf()` function was run with prior probabilities: $p_1 = p_2 = 10^{-4}$, $p_{12} = 5 \times 10{-6}$.

Using `coloc.bf_bf()`, was performed for overlapping GWAS and eQTL defined regions where there were credible sets for both GWAS and eQTL datasets. 'Low purity' credible sets present in the FinnGen data were removed. The `coloc.bf_bf()` function was run with default prior probabilities: $p_1 = p_2 = 10^{-4}$, $p_{12} = 5 \times 10^{-6}$.

### UK Biobank GWAS-eQTL analysis

We used a heuristic algorithm to define GWAS peaks, performing 5 rounds of iteratively merging peaks within 1MB of each other.

To perform finemapping with variant-specific priors on the GWAS data at each peak for each trait we used the `finemap.abf()` function in coloc. We used the uniform, precomputed PolyFun and trait-specific PolyFun priors in this analysis.

Using `coloc.abf()`, we performed colocalisation across all identified GWAS peaks for all three traits against 5 eQTL datasets in disease-relevant tissues per trait (Table C in S1 Text). Colocalisation was only performed if the region contained at least one genome-wide significant GWAS SNP and at least eQTL with p-value less than $5 \times 10^{-6}$. The `coloc.abf()` function was run with prior probabilities: $p_1 = p_2 = 10^{-4}$, $p_{12} = 5 \times 10^{-6}$. In this analysis we used the following precomputed PolyFun and trait-specific PolyFun priors applied to the GWAS data.

### Open Targets Genetics data

We downloaded the `latest` tranche of the Open Targets Genetics data from the Open Targets Genetics FTP site (https://ftp.ebi.ac.uk/pub/databases/opentargets/genetics/). We extracted the $\Pr(H_4)$ from all the colocalisation results present in the `v2d_coloc` folder and then filtered to $\Pr(H_4) > 0.5$. As analysing all variants was computationally impractical we randomly sampled 1,000,000 SuSiE results for variants from the `v2d_credset` folder. The variants were filtered to be the lead variant in each locus and the posterior inclusion probability (PIP) was extracted. The PIPs were further filtered to be >0.5.

## Acknowledgements

We thank members of the Wallace group and the MRC Biostatistics Unit for helpful discussions. We acknowledge the participants and investigators of the FinnGen study.

## Supporting Information

**S1 Text. Supplementary Methods and Tables**. Contains 5 pages of Supplementary Methods and Supplementary Tables A, B and C.
(PDF)

**S1 Fig. Benchmark of coloc execution speed with variant-specific and uniform priors**
Beeswarm plot of execution time of coloc with variant-specific and uniform priors. (**a**) Datasets with 500 variants. (**b**) Datasets with 2000 variants. Using variant-specific priors has no meaningful effect on speed, especially when the number of variants is small.
(TIFF)

**S2 Fig. Distribution of eQTL-TSS distance across datasets** Boxplots of distance (measured in base pairs, $\log_{10}$ scale) of eQTLs to TSS for genome-wide significant eQTLs in v6 of the eQTL catalogue (including GTEx v8), eQTLGen and OneK1K dataset. Each boxplot corresponds to a different study.
(TIFF)

**S3 Fig. Distribution of eQTL-TSS distance in the OneK1K dataset**. (**a**) Boxplots of distance (measured in base pairs) of eQTLs to TSS (log10 scale) by round of conditional analysis. (**b**) Boxplots of distance (measured in base pairs) of eQTLs to TSS ($\log_{10}$ scale) by cell type. (**c**) Boxplots of distance (measured in base pairs) of eQTLs to TSS (log10 scale) by the number of cell types the eQTL is significant in.
(TIFF)

**S4 Fig. Credible set and colocalisations posterior probabilities in OpenTargets Genetics data**. (**a**) Random sample of posterior inclusion probabilities from the lead variant in 95% credible sets calculated using SuSiE in the OpenTargets Genetics platform. (**b**) All $\Pr(H_4)$ calculated by coloc in the OpenTargets Genetics platform. Both plots are restricted to probabilities >0.5. (See Methods for details.)
(TIFF)

**S5 Fig. Impact of priors on posterior probabilities in the pQTL-eQTL comparison**. (**a**) Plot of the proportion of loci with absolute value change in $\Pr(H_4)$ less than 0.01 for coloc method with all priors. (**b**) Boxplot of maximum log Bayes factor calculated by coloc-single or calculated by SuSiE and used as input to coloc-susie across loci. (**c**) Scatter plot of maximum log Bayes factor for coloc-single method against the difference in uniform and variant-specific priors, taking the median over prior information sources. Each point is a locus. (**d**) Scatter plot of $\Pr(H_4)$ calculated with a uniform prior vs with an eQTLGen-estimated eQTL-TSS density prior across all tested pQTL-eQTL loci for which colocalisation was performed. The red lines show the 0.8 significance threshold for both coloc with uniform and variant-specific priors. (**e**) Scatter plot of $\Pr(H_4)$ with uniform priors vs the difference between $\Pr(H_4)$ with uniform and eQTLGen-estimated eQTL-TSS density prior across all pQTL-eQTL loci where colocalisation was performed. The grey lines show the maximum possible values of the

difference in colocalisation probabilities given the value of $Pr(H_4)$ calculated with uniform prior probabilities.
(TIFF)

**S6 Fig. Trait-specific prior and precomputed PolyFun calculated prior probabilities** The probabilities are displayed in a +/-500kB region around the canonical TSS, shown by the dashed black line, of the *VAV3* gene; the same region as in Fig 1.
(TIFF)

**S7 Fig. The impact of PolyFun and eQTLGen priors in the UK Biobank trait**. (**a**) Credible set size using different priors across three traits. (**b**) As in **??**, the effect of priors on colocalisation significance across three UK Biobank traits. Here we also consider applying the PolyFun priors to the GWAS dataset and eQTLGen prior to eQTLGen dataset simultaneously.
(TIFF)

**S1 Table. List of colocalisations with changed significance in colocalisation between FinnGen GWAS traits and 5 eQTL datasets per trait results for coloc with the single-causal variant assumption.**
(MS Excel)

**S2 Table. List of colocalisations with changed significance in colocalisation between FinnGen GWAS traits and 5 eQTL datasets per trait for coloc with SuSiE.**
(MS Excel)

## Author contributions

**Conceptualization:** Chris Wallace.

**Formal analysis:** Jeffrey M. Pullin.

**Funding acquisition:** Chris Wallace.

**Investigation:** Jeffrey M. Pullin, Chris Wallace.

**Methodology:** Jeffrey M. Pullin, Chris Wallace.

**Project administration:** Chris Wallace.

**Software:** Jeffrey M. Pullin, Chris Wallace.

**Supervision:** Chris Wallace.

**Validation:** Jeffrey M. Pullin.

**Visualization:** Jeffrey M. Pullin.

**Writing – original draft:** Jeffrey M. Pullin, Chris Wallace.

**Writing – review & editing:** Jeffrey M. Pullin, Chris Wallace.

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
