## [Decision Letter · Decision Letter 0]

8 Oct 2024

Dear Dr Pullin,

Thank you very much for submitting your Methods entitled 'Variant-specific priors in colocalisation analysis' to PLOS Genetics.

The manuscript was fully evaluated at the editorial level and by independent peer reviewers. The reviewers appreciated the attention to an important problem, but raised some substantial concerns about the current manuscript. Based on the reviews, we will not be able to accept this version of the manuscript, but we would be willing to review a much-revised version. We cannot, of course, promise publication at that time.

A major concern raised by both reviewers is the minimal impact of the proposed variant-specific prior, with the practical gains appearing incremental. This issue needs to be clearly addressed, and the advantages of the proposed method, along with its contributions, should be more convincingly demonstrated before the work can be further considered.

If you decide to revise the manuscript for further consideration at PLOS Genetics, please aim to resubmit within the next 60 days, unless it will take extra time to address the concerns of the reviewers, in which case we would appreciate an expected resubmission date by email to plosgenetics@plos.org.

To resubmit, log into your Editorial Manager account and select the option 'Revise Submission' in the 'Submissions Needing Revision' folder.

We are sorry that we cannot be more positive about your manuscript at this stage. Please do not hesitate to contact us if you have any concerns or questions.

Yours sincerely,

Lin Chen, Ph.D.

Academic Editor

PLOS Genetics

Michael Epstein

Section Editor

PLOS Genetics

Reviewer's Responses to Questions

**Comments to the Authors:**

Reviewer #1: Summary of the Paper:

The manuscript addresses the challenge of linking GWAS variants to their causal genes using the coloc method, a widely used approach for statistical colocalisation analysis. The authors propose incorporating variant-specific prior probabilities, based on diverse data sources such as PolyFun, non-coding constraint (Gnocchi), and enhancer-gene link scores (ABC), into coloc to improve performance. Their findings show that using these priors improves colocalisation accuracy in up to 13.5% of loci, demonstrating the benefits of integrating additional variant-level information.

Scientific Novelty:

The study is novel in extending the coloc method by introducing variant-specific prior probabilities to account for the heterogeneity of variant information, a concept well established in GWAS fine-mapping but not previously explored in colocalisation. This work bridges the gap between fine-mapping and colocalisation analysis, addressing the limitations of uniform priors in coloc. The systematic evaluation of multiple sources of variant information in real-world pQTL-eQTL and GWAS-eQTL colocalisation adds to the value of this contribution.

Major Concerns:

1. Limited improvement in both simulation and real data scenarios: Despite the potential of variant-specific priors, the reported improvement in performance (2-3% increase in recall and precision) is modest. The study highlights that the impact of these priors was often small, especially in GWAS-eQTL colocalisation. This may question the practical utility of the method for large-scale applications, where computational costs must be justified by substantial performance gains.

2. Lack of technical contribution: The author did not make attempt to improve the co-localization fine-mapping performance. For instance, the importance of the annotations may change depending on the traits. In PolyFun and SparsePro, the annotation weights are learned as *empirical priors*. But in the proposed approach the prior (while variant-specific) are fixed.

3. Lack of direct application of PolyFun to the GWAS traits of interest: The authors mention that, for computational reasons, PolyFun was not directly applied to the GWAS datasets analyzed, and pre-computed priors from UK Biobank traits were used instead. Given that PolyFun is dataset-specific, this compromises the applicability of these results to different traits, and the conclusions drawn may not fully reflect the method’s potential. Another alternative is SparsePro (Zhang et al., PloS Genetics 2024) and SharePro (Zhang et al., Bioinformatics 2024), which was published more recently and more efficient than PolyFun.

4. Generalizability to other GWAS contexts: The study largely focuses on pQTL-eQTL colocalisation with minimal testing on diverse GWAS traits. The results of colocalisation in autoimmune disease, type 2 diabetes, and hypertension in FinnGen are briefly mentioned, but the manuscript would benefit from further exploration and a broader comparison to other trait categories.

5. Absence of strong benchmarks for comparison: While the study uses pQTL-eQTL colocalisation as a ground truth benchmark, it remains unclear how the method compares against other cutting-edge colocalisation methods (e.g., SharePro) or how well it would perform in a setting with more complex polygenic traits. Introducing stronger benchmarks would improve the robustness of the results.

Minor Concerns:

1. Code and reproducibility: The paper references that the code for implementing the method is available in a GitHub repository. It would be beneficial to explicitly include a link in the manuscript and provide a minimal working example to allow users to apply the methods on their own data.

2. The first two sections under Results are mostly method description would be more appropriate under Methods section

3. Threshold choices: The manuscript uses a threshold of Pr(H4) > 0.8 to define colocalisation significance. It would be useful to provide additional context or references for why this threshold was chosen, particularly since other colocalisation studies may use different thresholds.

4. It would be helpful to have an overview Figure as Figure 1 to give a high-level conceptual diagram of the proposed method.

5. Supplementary material integration: There is heavy reliance on supplementary material for important methodological details. Integrating more of this content into the main manuscript, or at least summarizing key points, would improve readability and allow for a more thorough understanding without requiring extensive cross-referencing.

Reviewer #2: The goal of this manuscript is to improve current coloc approaches by adding weights to each variant based on heritabilities, functional annotation and eQTL-TSS weights. This is an interesting paper as coloc approaches are becoming very popular to integrate genetics of complex diseases with QTL for molecular traits.

The authors demonstrate that adding the weights (based on eQTLs, ABC, Gnocchi and PolyFun) the coloc results improve but only 2-3% in both simulated and real data. I think that adding the option “variant-specific-priors” to coloc is great, specially when people can add their own weights.

As mentioned the authors used several approaches to calculate the weights and perform the benchmarking. However there are several major assumptions in this manuscript that are not totally real and can lead to wrong inferences that need to be taken into account and also mention as limitations.

The first one is that the authors mention: “This analysis showed relatively little variation between tissues in GTEx or the eQTL catalogue, suggesting that tissue-specific densities are not needed.”, which I am not sure, I agree, as in Figure 1, there is not analyses by tissue. There are plenty of studies that indicate that many QTLs are tissue specific, and this may have an impact of the coloc results, if the weights are variant and tissue specific

The second is the “central dogma” in molecular biology of protein and mRNA levels are correlated, which is not clearly not true. The production of a protein is clearly associated with mRNA levels. But once the protein is produced, their stability in the cell, whether is secreted, cleavage …. does not depend on the mRNA levels. There are many studies that demonstrate that protein levels in humans does not correlate with mRNA levels. The authors also say that if there is a pQTL there needs to be a eQTL is also not true. Proteogenomic studies have demonstrated that pQTL are enriched for coding variants, likely affecting intracellular trafficking, membrane affinity, cleavage, protein binding among others. Those pQTL are not eQTL and not an artifact of the antibody or aptamer binding. Also pQTL are highly tissue specific, which connect to the previous critique. Therefore, using eQTLgen or OneK1K as a weights for pQTL will lead to wrong inferences. The inferences here will be lower in this scenario as both eQTL and pQTL are from blood, but it is not clear if this will be translated well for QTL generate in other tissues.

The other major limitation, not mentioned or discussed on the manuscript is that all the databases uses here are based on Non-Hispanic white datasets (I think OneK1K has some non-white datasets, but not clear how many and how was used). Different populations have different LD and different QTLs. Leverage other LD maps may have more impact on improving coloc results that incorporating weights. At least a discussion of this should be included on the manuscript.

**Have all data underlying the figures and results presented in the manuscript been provided?**

Reviewer #1: Yes

Reviewer #2: Yes

PLOS authors have the option to publish the peer review history of their article (what does this mean?). If published, this will include your full peer review and any attached files.

Reviewer #1: No

Reviewer #2: No

---

## [Decision Letter · Decision Letter 1]

26 Mar 2025

PGENETICS-D-24-00952R1

Variant-specific priors clarify colocalisation analysis

PLOS Genetics

Dear Dr. Pullin,

Thank you for submitting your manuscript to PLOS Genetics. The revised manuscript is improved over the previous submission. However, one reviewer still have some concerns regarding the interpretation of the results.  Therefore, we invite you to submit a revised version of the manuscript that addresses the points raised.

Please submit your revised manuscript within 30 days . If you will need more time than this to complete your revisions, please reply to this message or contact the journal office at plosgenetics@plos.org. Please include the following items when submitting your revised manuscript:

We look forward to receiving your revised manuscript.

Kind regards,

Lin Chen, Ph.D.

Academic Editor

PLOS Genetics

Michael Epstein

Section Editor

PLOS Genetics

Aimée Dudley

Editor-in-Chief

PLOS Genetics

Anne Goriely

Editor-in-Chief

PLOS Genetics

**Journal Requirements:**

1) Please provide an Author Summary. This should appear in your manuscript between the Abstract (if applicable) and the Introduction, and should be 150-200 words long. The aim should be to make your findings accessible to a wide audience that includes both scientists and non-scientists. Sample summaries can be found on our website under Submission Guidelines:

https://journals.plos.org/plosgenetics/s/submission-guidelines#loc-parts-of-a-submission

2) Please ensure that your article adheres to the standard Methods article layout and order of Abstract, Author Summary, Introduction, Description of the Method, Verification and Comparison, Applications, Discussion, Acknowledgements, References, and Supplementary Information. For details on what each section should contain, see our Methods article guidelines:

https://journals.plos.org/plosgenetics/s/submission-guidelines#loc-manuscript-organization.

**Reviewers' comments:**

Reviewer's Responses to Questions

Reviewer #1: All my comments are addressed.

Reviewer #2: This is a revised manuscript focused on improving coloc approaches by adding variant specific weights.

My previous concerns were related to having weights based only on eQTL and using only EUR-specific weights. The authors have addressed my comments related with the European-specific weights.

I still do fully not agree with their justification for the pQTL. The authors indeed indicate that only 65% of the pQTL colocalize with eQTL, so a large number of then, 35% do not. And most of this pQTLs that do not colocalize are due to posttranslational events. I think that talking here about the central dogma, is indeed, simplifying biology too much that is misleading. Proteins levels are regulated at many more levels than mRNA, and proteins, a and not RNA, are normally the effector molecules in diseases and biological traits. Saying that mRNA -related data can just be translated into protein data is one of the reasons so many loci are still no solved. I would suggest removing the wording related the central dogma and that the pQTLs are expected to be eQTL as this is clearly erroneous and misleading.

It would be ok with me, if the authors just have a few sentences in the discussion that including tissue-specific and other molecular traits QTL can improve the coloc. The can add it to the sentence where they mention the users can include their own weight.

**Have all data underlying the figures and results presented in the manuscript been provided?**

Reviewer #1: None

Reviewer #2: Yes

PLOS authors have the option to publish the peer review history of their article (what does this mean?). If published, this will include your full peer review and any attached files.

Reviewer #1: No

Reviewer #2: No

**Figure resubmission:**
---

## [Editor Report · Decision Letter 2]

21 Apr 2025

Dear Dr Pullin,

We are pleased to inform you that your manuscript entitled "Variant-specific priors clarify colocalisation analysis" has been editorially accepted for publication in PLOS Genetics. Congratulations!

Yours sincerely,

Lin Chen, Ph.D.

Academic Editor

PLOS Genetics

Michael Epstein

Section Editor

PLOS Genetics

Aimée Dudley

Editor-in-Chief

PLOS Genetics

Anne Goriely

Editor-in-Chief

PLOS Genetics

Comments from the reviewers (if applicable):

**Data Deposition**

http://datadryad.org/submit?journalID=pgenetics&manu=PGENETICS-D-24-00952R2

**Press Queries**

---

## [Editor Report · Acceptance letter]

PGENETICS-D-24-00952R2

Variant-specific priors clarify colocalisation analysis

Dear Dr Pullin,

We are pleased to inform you that your manuscript entitled "Variant-specific priors clarify colocalisation analysis" has been formally accepted for publication in PLOS Genetics! Your manuscript is now with our production department and you will be notified of the publication date in due course.

With kind regards,

Anita Estes

PLOS Genetics

On behalf of:
